
# The Four-Wavelength Photoacoustic Aerosol Absorption Spectrometer PAAS-4λ

Franz Martin Schnaiter[1,2], Claudia Linke[1], Eija Asmi[3], Henri Servomaa[3], Antti-Pekka Hyvärinen[3], Sho Ohata[4,5], Yutaka Kondo[6], and Emma Järvinen[1]

[1]Karlsruhe Institute of Technology, Kaiserstraße 12, 76021 Karlsruhe, Germany
[2]schnaiTEC GmbH, Werner-von-Siemens-Str. 2-6, 76646 Bruchsal, Germany
[3]Finnish Meteorological Institute, P.O. Box 503, 00101 Helsinki, Finland
[4]Institute for Space–Earth Environmental Research, Nagoya University, Nagoya, Aichi, Japan
[5]Institute for Advanced Research, Nagoya University, Nagoya, Aichi, Japan
[6]National Institute of Polar Research, Tokyo, Japan

**Correspondence:** Martin Schnaiter (martin.schnaiter@kit.edu)

**Abstract.**

In this paper the Photoacoustic Aerosol Absorption Spectrometer PAAS-4λ is introduced. PAAS-4λ was specifically developed for long-term monitoring tasks in (unattended) air quality stations. It uses four wavelengths coupled to a single acoustic resonator in a compact and robust set-up. The instrument has been thoroughly characterized and carefully calibrated in the laboratory using $NO_2$/air mixtures and Nigrosin aerosol. It has an ultimate $1\sigma$ detection limit below 0.1 $Mm^{-1}$ at a measurement precision and accuracy of 3% and 10%, respectively. In order to demonstrate the PAAS-4λ suitability for long-term monitoring tasks, the instrument is currently validated at the air quality monitoring station Pallas in Finland, about 140 km north of the Arctic circle. Eleven months of PAAS-4λ data from this deployment are presented and discussed in terms of instrument performance. Intercomparisons with the filter-based photometers COSMOS, MAAP, and AE33 demonstrate the capabilities and value of PAAS-4λ, also for the validation of the widely used filter-based instruments.

## 1 Introduction

Black carbon (BC) particles from combustion emissions (i.e. soot) are commonly monitored by filter-based methods like Aethalometers®. In these methods, the ambient air is sampled through quartz fibre filters where aerosol particles are deposited in the filter matrix. The attenuation of light by absorption from dark particles embedded in the filter is then continuously measured across the filter thickness while the aerosol is sampled. The correlation between the measured attenuation coefficient $b_{ATN}$ of the filter-embedded particles to the absorption coefficient $b_{abs}$ of the particles in airborne state is derived from specific calibration procedures (Weingartner et al., 2003). A fundamental problem with such filter-based methods is that the filter matrix also interacts with the light through multiple light scattering, which increases the light path within the filter matrix by an unknown factor. As a consequence the light absorbing particles deposited in the filter matrix are participating many times in the light absorption process, which results in an increase of $b_{ATN}$ with respect to $b_{abs}$. The light attenuation by the particle loaded filter is further affected by the amount of particles deposited in the filter matrix as well as their optical properties by the

 

so-called loading or shadowing effect. Here, the accumulated particle mass (typically on one side of the filter) can result in a total blocking of light portions, so that particle mass located deeper in the filter gets relatively less light. This leads in general to a decrease of $b_{\mathrm{ATN}}$ with respect to $b_{\mathrm{abs}}$. However, this effect might be partially counterbalanced by any light scattering

particle mass that is co-deposited in the filter matrix. Therefore, the raw attenuation measurements have to be corrected for the above effects by introducing the correction factors $C$ and $R(\mathrm{ATN})$ for the multiple-scattering and loading effect, respectively (Weingartner et al., 2003).

There are many studies investigating these effects and providing correction schemes for $C$ and $R(\mathrm{ATN})$. For example, the recent study by Luoma et al. (2021) compared different correction algorithms for a several years data set from a boreal forest

site in Finland. The general result of this study is that the multiple-scattering correction factor $C_{\mathrm{ref}}$ for the Aethalometer® AE31 (Magee Scientific Co., USA) varies between 3.09 and 3.34 (median value) among the different correction schemes but with a total variability between 2.23 and 4.26 (5 and 95 % percentiles). Thus, uncertainties of nearly 100% can occur in hourly averaged data depending most likely on the actual variability in aerosol composition, size, and concentration (i.e. pollution level). The term $C_{\mathrm{ref}}$ indicates that the $C$ value was deduced by using the $b_{\mathrm{abs}}$ measurements from a collocated Multi-Angle

Absorption Photometer (MAAP, Thermo Fisher Scientific, USA, model 5012, discontinued) as reference. It has to be noted that the MAAP also represents a filter-based method, but consists of a separate measurement of the diffuse light-scattering from the particle-loaded filter matrix (Petzold and Schönlinner, 2004), which has been proven to be less prone to filter-induced artefacts even under high aerosol light scattering contributions (e. g. Schnaiter et al., 2005). The finding of highly variable $C$ values at the Finnish boreal site is supported by reported $C_{\mathrm{ref}}$ factors from different locations that indicate higher $C$ values for

sites that are influenced by higher pollution levels (Collaud Coen et al., 2010). Further, Luoma et al. (2021) concluded that the correction factors $C$ and $R(\mathrm{ATN})$ are most likely also wavelength dependent to an unknown extent. This has consequences for the reliability of the deduced wavelength dependence of $b_{\mathrm{abs}}$, i.e. the Absorption Ångström Exponent (AAE), which is frequently used to perform a source apportionment of the aerosol in terms of fossil fuel combustion or biomass burning sources. In order to deduce the spectral absorption coefficient $b_{\mathrm{abs}}(\lambda)$ more reliably and independent of the level and type of pollution,

non-intrusive methods like photoacoustic spectroscopy (PAS) or photo-thermal interferometry (e.g., Visser et al., 2020) have to be applied (Moosmüller et al., 2009). Having such instruments with sufficient robustness, stability, and sensitivity available also for a long-term deployments at (remote) field sites, would be ideal not only for highly accurate measurements of $b_{\mathrm{abs}}$ and AAE, but also for further investigations of the above uncertainties of the filter-based instruments that are the most commonly applied at long-term measurements at different sites (Luoma et al., 2021).

PAS for aerosol research has been greatly improved within the last decade mainly as a consequence of large development steps in laser technology. Several PAS instruments have been developed in different research labs that show low detection levels and measure at several wavelengths (Lewis et al., 2008; Ajtai et al., 2010; Haisch et al., 2012; Lack et al., 2012; Sharma et al., 2013; Linke et al., 2016; Fischer and Smith, 2018; Foster et al., 2019). Instruments that comprise several wavelengths coupled to a single acoustic resonator are the most promising concepts in terms of future operations in air monitoring stations.

Up to now, these state-of-the-art instruments have mainly been used in lab studies or in dedicated field projects but never in remote monitoring stations. The reasons might be that these instruments still need considerable maintenance to keep them





operational or are tight to specific research projects. A commercial three-wavelength PAS instrument became available by the beginning of the 2010s (PASS-3, Droplet Measurement Technologies, USA), that was an extension of the two-wavelength prototype versions originally developed at the Desert Research Institute, Reno (Lewis et al., 2008). However, this instrument

was discontinued later.

In this paper the Four-Wavelength Photoacoustic Aerosol Absorption Spectrometer PAAS-4$\lambda$ is introduced. The PAAS-4$\lambda$ is based on a prototype version developed at KIT (Linke et al., 2016), but has been significantly improved in size, sensitivity, stability, and operability through the research spin-off schnaiTEC. Recently, schnaiTEC has started to market the PAAS-4$\lambda$ with the goal to make this state-of-the-art PAS technology available for laboratory research, air quality studies, and long-term

monitoring tasks at remote air quality stations. To reach these goals the PAAS-4$\lambda$ instrument is currently operated at the Finnish air quality monitoring station Pallas located about 140 km north of the Arctic circle. Data from this deployment are shown here to demonstrate the PAAS-4$\lambda$ functionality in this target operational environment. The design and operation principle of PAAS-4$\lambda$ is presented in Sect. 2 followed by a description of the instrument characterization and calibration procedures in Sect. 3. Measurement examples from the long-term deployment at the Pallas remote field site are presented and discussed in Sect. 4. A

summary and an outlook is given in Sect. 5.

## 2   Instrument Design and Operation Principle

The fundamental operation principle of photoacoustic (PA) systems for gas as well as aerosol detection including PA cell design considerations is reviewed elsewhere (Rosencwaig, 1980; Miklós et al., 2001; Bozóki et al., 2011; Haisch, 2012). Here only those aspects of the PA theory and instrument design are described that are relevant for the PAAS-4$\lambda$. The PA signal

is generated in an aerosol consisting of absorbing particles by illuminating the aerosol with modulated incident (laser) light having a modulation frequency of $f_m$. By absorbing the light in the on-phase of the modulation the light absorbing (dark) particles heat up while the non-absorbing (white) particles do not. In the subsequent off-phase of the modulation the deposited heat is released to the surrounding gas resulting in a local pressure change. As the incident light is continuously modulated the induced pressure fluctuations are also periodic resulting in a sound wave with frequency $f_m$ that is expanding in the aerosol. If

the PA signal generation is conducted in an acoustic resonator and if the modulation frequency of the incident light is tuned to one of the acoustic modes of this resonator the sound wave is amplified. A sensitive microphone attached to the PA cell then detects and converts the sound pressure to a voltage. The generated PA signal can be expressed as following (Ajtai et al., 2010; Bozóki et al., 2011)

$$S = P_0 \cdot M \cdot (C_{\text{cell}} \cdot \sigma_{\text{abs}} \cdot c + A_{\text{b}}), \tag{1}$$

with $S$ the PA signal in [V], $P_0$ the Fourier component of the incident light power in [W], $M$ the sensitivity of the microphone in [V·Pa$^{-1}$], $C_{\text{cell}}$ the PA cell constant in [Pa·m·W$^{-1}$], $\sigma_{\text{abs}}$ the absorption cross section of the particles in [m$^2$], $c$ the number concentration of the absorbing particles in [m$^{-3}$], and $A_{\text{b}}$ the background signal in [Pa·W$^{-1}$]. As the aerosol is in general composed of different absorbing particle species with different absorption cross sections $\sigma_{\text{abs}}$, the term $\sigma_{\text{abs}} \cdot c$ in Eq. (1) is





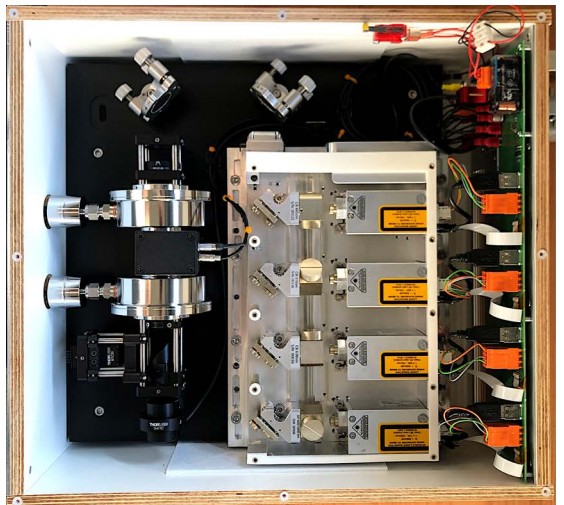
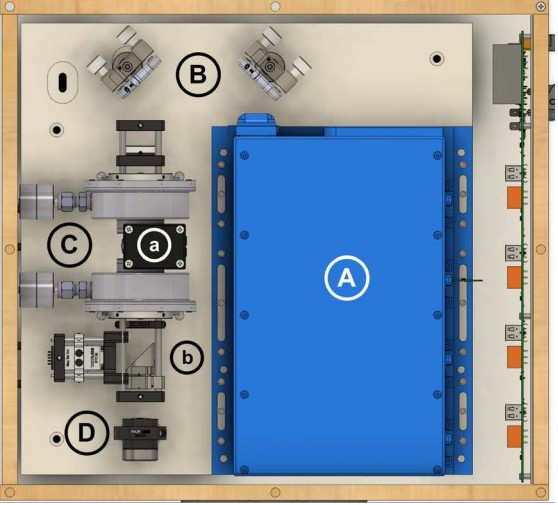

**Figure 1.** The optics unit of PAAS-4$\lambda$. (Left) Photo with laser combiner lid opened. In this setup the laser combiner hosts four lasers. (Right) Design model with labels of the main components; laser combiner (A), beamsteering mirrors (B), PA cell (C) with microphone unit (a) and beamsplitter optics (b), and laser power meter (D).

replaced by $b_{\mathrm{abs}}$ the total absorption coefficient of the aerosol in $[\mathrm{m}^{-1}]$. Hence, the physical quantity measured by PAAS-4$\lambda$ is

the absorption coefficient $b_{\mathrm{abs}}$.

Following the above operation principle a photoacoustic spectrometer (PAS) basically consists of (i) modulated lasers, (ii) a photoacoustic cell equipped with microphone and laser power measurement, and (iii) electronics for control and signal filtering. These components are presented more in detail in the following sections for the PAAS-4$\lambda$ system.

## 2.1 Optics Unit

The optics unit of PAAS-4$\lambda$ hosts the lasers, the photoacoustic cell, and the laser power meter (Fig. 1). These components are set-up on a 13 mm thick Aluminum base plate with 380 mm × 380 mm edge length. The optics base plate is supported by four 1.5 inch in diameter stainless steel posts each equipped with a 1 inch thick Sorbothane foot (AV5, Thorlabs Inc., USA) for passive vibration insulation. A 462 mm × 422 mm × 200 mm (L×B×H) birch plywood housing with 13 mm panel thickness hosts the optics set-up. The housing is equipped with a laser interlock switch and, thus, represents a class 1 laser enclosure

for eye and skin safe routine operation without any laser-safety precautions necessary at the deployment site. Two ventilators implemented in the front panel in conjunction with a passive heat sink underneath the optics base plate ensures sufficient release of the heat dissipated by the lasers into the base plate. This results in stable thermal conditions inside the optics unit with an equilibrium temperature of about 30 °C.





### 2.1.1 Laser System

The lasers are implemented in a beam combiner that can host up to four lasers (LightHUB®-4, Omicron-Laserage Laserproduke GmbH, Rodgau-Dudenhofen, Germany). Diode lasers with different emission wavelengths are used in the laser combiner according to the user needs (LuxX®+, Omicron-Laserage Laserproduke GmbH, Rodgau-Dudenhofen, Germany). The wavelength range that can be covered by these lasers is from 375 nm in the near-UV to 1550 nm in the near-IR. The lasers can be digitally modulated over a wide frequency range up to 250 MHz, they have a circular Gaussian beam profile with a specified

$(1/e^2)$ diameter of $1.25 \pm 0.25$ mm, a beam ellipticity of better than 1.1:1, a vertical polarisation ratio of $> 100:1$, and a long term power stability below 0.5% over 8 h. The laser diode is temperature stabilised to 25 °C by a Peltier device dissipating heat to the base plate of approximately 5 to 10 W. Each individual laser in the LightHUB® consists of a beam shifter and a beam combiner. The beam combiners are equipped with specific dichroic filters according to the actual laser wavelengths that are used in the system. The LightHUB® optics allows a stable alignment of the up to four laser beams on a single optical

axis. Two beam steering mirrors with broadband anti-reflection coating (e.g., BB1-E02, Thorlabs Inc., USA, for the 400 nm to 750 nm wavelength range) are used to overlap the combined laser beam with the optical axis of the PA cell. These mirrors are aligned with two Polaris® low-drift kinematic mounts (POLARIS-K1, Thorlabs Inc., USA) which provide excellent long-term alignment stability of less than 2 μrad after a thermal cycling of 12°C.

### 2.1.2 Photoacoustic Cell

Figure 2 shows a cross-section of the photoacoustic cell of the PAAS-4$\lambda$ system. The PA cell basically represents a resonant longitudinal open acoustic cavity. The cavity comprises a tube of 6.5 mm diameter and 49 mm length that is drilled in a 16 mm diameter cylindrical stainless steel body. The tube walls are polished to a residual roughness of less than $R_a = 0.6$ μm. The cavity consists of a 4 mm diameter hole positioned at the half length of the tube, i.e. at the position of the anti-node of the fundamental longitudinal mode, that will host the microphone duct of 1.35 mm diameter. The cavity has the longitudinal

acoustic modes

$$f_n = \frac{n \cdot c}{2(l + \Delta l)} \quad n = 1, 2, 3, ..., \tag{2}$$

where $l$ is the length of the cavity and $c$ the speed of sound. The term $\Delta l$ has to be added to each open cavity end to take into account the mismatch between the one-dimensional field inside the cavity and the three-dimensional field outside (Miklós et al., 2001). The PAAS-4$\lambda$ acoustic resonator thus has a nominal frequency of $f_1 = 3240$ Hz assuming a speed of sound of

$c = 343$ m/s (at 20°C), a cavity length of $l = 49$ mm, and an end correction length $\Delta l \approx 0.61r$, with $r = 3.25$ mm the cavity radius. Note from Eq. (2) that $f_1$ depends on the temperature of the air inside the cavity since the speed of sound is mainly temperature dependent. This gives a frequency shift of about 6 Hz per °C, which has to be taken into account when operating the PAAS-4$\lambda$ system under rough thermal conditions where the temperature inside the optics unit is unstable and varies by more than 5 to 10 °C. See Figure S4 in the Supplement for a frequency scan of the fundamental longitudinal mode $f_1$ of the

PA cell.





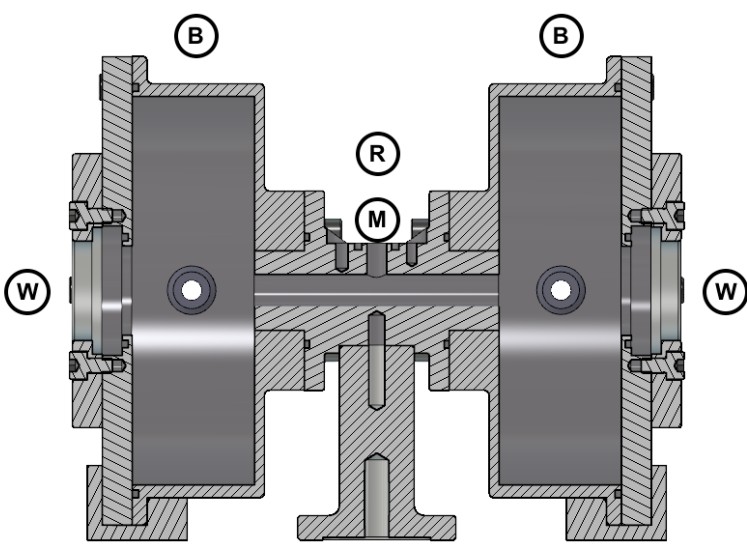

**Figure 2.** The photoacoustic cell of PAAS-4$\lambda$ with the following main components. $\lambda/2$ acoustic resonator (R), $\lambda/4$ acoustic buffers (B), laser entrance and exit windows (W). The position of the microphone (not shown) is indicated by (M).

The $6.5\,\mathrm{mm}$ diameter acoustic cavity is steplessly expanded to $78\,\mathrm{mm}$ diameter at both ends where two $\lambda/4$ acoustic buffers are mounted (Fig. 2). These buffers comprise the aerosol inlet and outlet ports as well as the laser entrance and exit windows (WL11050, Thorlabs Inc., USA) that complete the PA cell. The buffers are used to simulate open cavity end conditions as well as to suppress ambient acoustic noise around the cavity resonance frequency (Bozóki et al., 2011). Further $\lambda/4$ acoustic buffers are connected to the inlet and outlet aerosol flow ports (located next to the (C) label in Fig. 1) to suppress acoustic noise in the flow system (e. g. generated by a turbulent flow or by the pump system).

The cell constant in Eq. (1) can be calculated from the specifications of the acoustic resonator (Bozóki et al., 2011).

$$C_{\mathrm{cell}} = \frac{(\gamma - 1) \cdot Q \cdot G}{2 \cdot \pi \cdot f_m \cdot A_{\mathrm{res}}}, \tag{3}$$

with $\gamma$ the ratio of specific heat constants, $f_m$ the modulation frequency in [Hz] that is adapted to the resonance frequency $f_1$ of the cavity, $A_{\mathrm{res}}$ the cross sectional area of the resonator in [m$^2$], $Q$ the quality factor of the excited acoustic mode, and $G$ the so-called geometric factor that describes the spatial overlap between the laser beam and the acoustic eigenmode of the resonator. For a well designed PA cell the geometric factor is close to unity. With the resonator diameter of $d_{\mathrm{res}} = 6.5 \cdot 10^{-3}\mathrm{m}$, the fundamental longitudinal mode frequency of $f_1 = 3240\,\mathrm{Hz}$, and the resonator quality factor of $Q = 21$ a cell constant of $C_{\mathrm{cell}} = 12.4\,\mathrm{Pa} \cdot \mathrm{m} \cdot \mathrm{W}^{-1}$ results. The used microphone (Knowles, model EK-3029) has a manufacturer specified sensitivity at the frequency $f_1 = 3240\,\mathrm{Hz}$ of about -50dB (relative to $1.0\mathrm{V}/0.1\mathrm{Pa}$), which corresponds to a microphone sensitivity $M =$



$0.032 \ \mathrm{V \cdot Pa^{-1}}$. Taking into account the microphone preamplifier gain of about $580$ and the lock-in amplifier gain of $30$ (typically used in the calibration procedure), a cell constant of about $C_{\mathrm{cell}} = 6900 \ \mathrm{V \cdot m \cdot W^{-1}}$ is calculated for the PAAS-4$\lambda$ cell. Here it has to be noted that the actual values of the microphone sensitivity $M$, the microphone preamplifier gain, the quality factor $Q$, as well as the geometric overlap factor $G$ can significantly vary from unit to unit and, therefore, a characterisation and calibration procedure is required to deduce the cell constant more reliably for the actual unit (see Sect. 3.1).

The laser power $P_0$ of the modulated laser is measured behind the PA cell exit window with a photodiode power sensor (S121C, Thorlabs Inc., USA). As it turned out that backscattered light from the diffusing element of the S121C sensor is producing a measurable increase of the PA signal background, beamsplitting optics is implemented in the exit laser path consisting of an iris and a 7:30 (R:T) beamsplitter plate (BST10, Thorlabs Inc., USA) that is reflecting approximately $80$ to $90\%$ of the laser beam power into a beam dump (BTC30, Thorlabs Inc., USA). In this way the diffuse backscattering is efficiently suppressed.

## 2.2 Electronics Unit

The electronics unit of PAAS-4$\lambda$ is implemented in a $19 \ \mathrm{inch}$ rack enclosure (INS 48680-L, Bopla GmbH, Germany) with a height of $310 \ \mathrm{mm}$, a width of $449 \ \mathrm{mm}$, and a depth of $495.5 \ \mathrm{mm}$ as part of the three-unit modular instrument set-up (see Fig. S1 of the Supplement for a photograph of the complete system). The unit comprises the following three major components (Fig. S2): the embedded real time controller (cRIO-9063, NI Corp., USA), the dual phase lock-in amplifier (LIA-MVD-200-L, Femto GmbH, Germany), and the touch screen panel computer (Panelmaster 0881, ICO GmbH, Germany). The embedded controller hosts four interface modules: two digital I/O boards (NI-9402) to provide the laser modulation and the lock-in reference frequency signals, one analog input board (NI-9205) to acquire the lock-in amplifier output voltages as well as the output from the temperature and humidity sensor, and a relay board (NI-9485) to provide switches for controlling peripheral equipment like the zero air filter bypass that is implemented in a separate flow unit (Fig. S1). The FPGA module of the cRIO-9063 in conjunction with one of the two digital I/O boards is used as a four channel Transistor-Transistor Logic (TTL) frequency generator with user adjustable frequency and duty cycle that drives the modulation of the lasers. This frequency generator is implemented with a bandwidth of 40 MHz, which gives a resolution of 1 Hz at the modulation frequency. The modulation frequency of the lasers can be independently set, which allows a simultaneous operation of all lasers at different frequencies and duty cycles. However, as the current set-up hosts only one lock-in amplifier, the lasers are usually operated in a sequential measurement procedure with the same frequency and duty cycle. A simultaneous measurement procedure with slightly different laser frequencies is possible in future set-ups that hosts separate lock-in amplifiers for each individual laser.

The preamplified microphone signal is analysed by a dual phase lock-in amplifier using the TTL laser modulation signal as reference. The lock-in amplifier converts (demodulates) the AC signal from the microphone to an amplified DC signal at the output while effectively suppressing the noise with a bandwidth adjustable low-pass filter. The LIA-MVD-200-L lock-in amplifier has the three different modes of operation, "High Dynamic Reserve", "Low Drift", and "Ultra Stable". We use the "Ultra Stable" mode in the PAAS-4$\lambda$ as this provides an excellent low-drift output signal with a DC-drift of only 5 ppm/K at a still sufficient dynamic reserve of 15 dB. The PAAS pre-amplified background signal is typically in the order of 100 μV





with a noise level of about 10 mV rms (see Fig. S3). Even at this relatively high noise level, lock-in gain settings up to 3,000 are within the dynamic reserve of the "Ultra Stable" mode without overloading the amplifier. The time constant of the low pass-filter is set to 1 s or 3 s for typical long-term measurements where the PAAS signal is averaged over several minutes. The 3 dB bandwidth of the low-pass filter is very narrow with 0.16 Hz or 0.05 Hz for a time constant of 1 s or 3 s, respectively. To further improve the noise suppression a second order low-pass filter with a roll-off slope of -12 dB/octave is applied in the

PAAS-4$\lambda$ lock-in amplifier.

The electronics unit comprises a touch screen panel computer for experiment control, data acquisition, and data storage. A graphical user interface (GUI) was designed in LabVIEW (NI Corp., USA) that fits to the $800 \times 600$ pixel resolution touch screen and that can be controlled without keyboard and mouse. One part of the PAAS LabView software package is installed on the cRIO-9063 real time controller that is feeding the FPGA with the frequency generator code. Further, this cRIO application

acts as middleware between the FPGA (connected to the digital output boards), the analog input board, and the relay board of the cRIO embedded system and the GUI application of the panel computer. In this way the time critical tasks "modulation frequency generation" and "data acquisition" are autonomously running on the real time embedded system which is separated from the Windows GUI application on the panel computer. Measurement data is then simply requested from the cRIO application according to the presets given by the user through the GUI application. The basic concept of the GUI application is to

autonomously operate the PAAS-4$\lambda$ through predefined measurement sequences. A script-based sequence language has been specifically developed that provides the sequence steps "Laser", "Measure", "Relay", "Repeat", "Scan", and "Wait", which can be arbitrarily combined and arranged in a measurement sequence by the user. Each sequence step has a set of keys that define the measurement or control tasks that need to be conducted within the step. The step "Measure", for example, has the keys "NumberOfPoints" and "DistanceBetweenPoints[timeConst]", which define how many samples (points) are averaged for

a stored measurement and at what time difference these samples are taken. Here, the time difference is defined as integer multiples of the lock-in amplifier time constant because the low-pass filter conducts exponential moving average in the time domain. The step "Scan" allows the autonomous performance of a frequency scan at predefined times, e. g., to correct the cell resonance frequency for temperature drifts. With the step "Relay" the user has access to the relay board and, thus, can implement peripheral equipment into the measurement sequence. This is primarily used for purging filtered air in background measurement

cycles, but controlling, e.g., inlet switches, denuder bypasses, or calibration gas provisions is also possible. This concept of predefined measurement sequences is an important prerequisite for any unattended long-term deployment of PAAS-4$\lambda$.

## 3  Instrument Calibration and Performance

### 3.1  Calibration

The PAAS-4$\lambda$ is calibrated using premixed $NO_2$/synthetic air calibration gas bottles (Air Liquide GmbH, CRYSTAL gas

mixture, $2\sigma$ maximum relative uncertainty: $\pm 5\%$). Different concentrations are used here with nominal volume mixing ratios of 200 ppb, 1000 ppb, and 5000 ppb. $NO_2$ is widely used as calibration gas for aerosol PAS applications (e. g., Arnott et al., 2000; Fischer and Smith, 2018) because the molecular absorption cross section is well known (Vandaele et al., 2002).





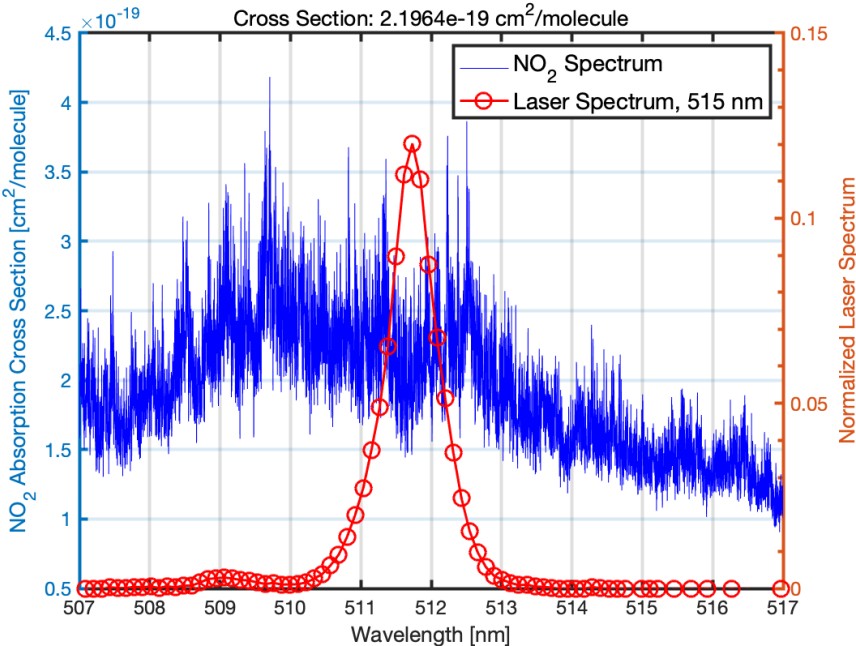

**Figure 3.** Normalized laser emission spectrum of the 515 nm LuxX®+ laser implemented in the PAAS-4$\lambda$-02-005 unit (red). The high resolution $NO_2$ molecular absorption cross section from Vandaele et al. (2002) is depicted in blue. A laser-specific absorption cross section of $2.23 \cdot 10^{-19}$ cm$^2$/molecule is deduced from the convolution of both spectra.

However, as the $NO_2$ spectrum is strongly varying across the visible spectrum, the calibration requires the knowledge of the $NO_2$ absorption cross section that is relevant for the specific PAAS-4$\lambda$ laser emission line. Therefore, the emission spectrum of
each individual laser unit is measured prior the implementation in PAAS-4$\lambda$ using a compact Czerny-Turner CCD spectrometer (CCS100/M, Thorlabs Inc., USA) with spectral accuracy of better than 0.5 nm at a spectral range of 350 nm to 700 nm. Figure 3 shows an example of the emission spectrum measured for a PAAS-4$\lambda$ LuxX®+ laser unit with a nominal emission wavelength of 515 nm. The high resolution molecular absorption cross section of $NO_2$, shown in blue in Fig. 3, has a significant variation from about 1.2 to $3.2 \cdot 10^{-19}$ cm$^2$/molecule in a 10 nm spectral window around the actual laser emission. The measured
laser emission spectrum is therefore mathematically convolved with the molecular $NO_2$ absorption cross section to deduce the laser-specific absorption cross section. In this example, a systematic bias of about 40% is generated in the calibration when using the manufacturer nominal emission wavelength, i.e., 515 nm, instead of the measured emission wavelength spectrum of the laser unit. The emission spectra measured for the 405 nm and 660 nm lasers of the corresponding PAAS-4$\lambda$ unit are given in the Supplement (Fig. S5) together with the emission spectra measured for another PAAS-4$\lambda$ unit.

In the calibration procedure premixed calibration gas is step-wise diluted with synthetic air by mixing the two gas flows in a specifically designed turbulent mixing chamber. The total flow is always set to 5 SLPM using two mass flow controllers. From this $NO_2$/synthetic air flow, PAAS-4$\lambda$ is sampling with 1 SLPM. The calibration procedure always starts with the undiluted





premixed bottle concentration to get the cell surfaces into equilibrium with the calibration gas (Fig. S6). The bottle $NO_2$ mixing ratio is then measured in 5 dilution steps 1:5, 2:5, 3:5, 4:5, and undiluted 5:5. At each dilution step the instrument measures

4 times 20 s periods of the lock-in R output for each laser in a cycle with 10 s waiting time after each laser switch. The individual measurements per dilution step are then averaged and corrected for the background signal from pure synthetic air that is measured before and after the calibration procedure. With the above initial bottle gas concentrations, this calibration procedure spans an $NO_2$ mixing ratio from 200 ppb to 5000 ppb which corresponds to more than 3 orders of magnitude in $NO_2$ absorption coefficient; from about $3 \cdot 10^{-6}$ m$^{-1}$ at a wavelength of 660 nm to $7 \cdot 10^{-3}$ m$^{-1}$ at a wavelength of 405 nm,

respectively. Figure 4 clearly demonstrates that the PA cell is represented by a single calibration constant which in turn means that the individual laser beams are well aligned to one axis and that their beam profiles are similar; otherwise the geometric overlap factor $G$ of Eq. (3) would be different for the different lasers.

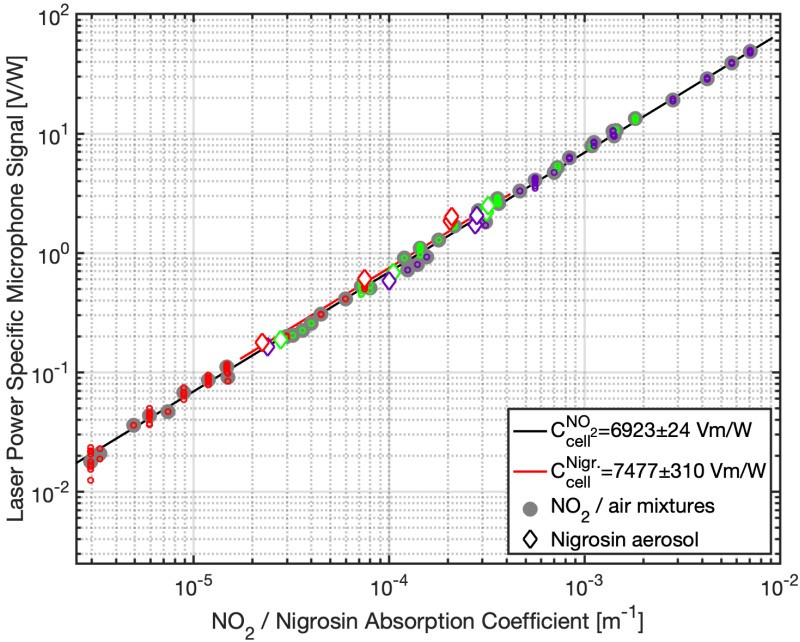

**Figure 4.** Calibration of a PAAS-4$\lambda$ unit hosting 405 nm (purple), 515 nm (green), and 660 nm (red) lasers. $NO_2$/synthetic air mixtures were presented to the instrument in a specific step-wise procedure (open circles). The cell constant of $6923 \pm 24$Vm/W is represented by the slope coefficient $\pm 1\sigma$ of the linear regression fit to the averaged data per $NO_2$ concentration (black line and bold grey circles). The data show a correlation coefficient of $R^2 = 0.998$ with the linear fit model. The calibration is verified using size selected Nigrosin aerosol particles (diamond symbols). A cell constant of $7477 \pm 310$Vm/W is deduced for the Nigrosin data only (red line). The two independently deduced cell constants differ by 8%. See text for details.

It is important to note here that the $NO_2$ absorption measurements at the 405 nm wavelength are affected by $NO_2$ photolysis (Roehl et al., 1994). Knowing the actual laser emission spectrum, the quantum yield of the $NO_2$ photolysis, and the $O_2+O\rightarrow$





$O_3$ recombination energy, the reduction of PA signal by $NO_2$ photolysis can be calculated and corrected. A correction factor of 1.3 is applied in Fig. 4 for the 405 nm measurements. A linear regression of the $NO_2$ deduces a regression slope, i.e., the cell constant $C_{\text{cell}}^{\text{NO}_2} = 6923$ V·m·W$^{-1}$, which is in a very good agreement with the theoretical value of 6900 V·m·W$^{-1}$ given in Sec. 2.1.2. The low relative uncertainty of 0.35% of the regression analysis is a result of the intrinsic high dynamic range and the wide linearity of PAS systems for gas measurements (Bluvshtein et al., 2017). An instrument precision of 3%

is deduced from day-to-day calibrations using the same bottle mixture and applying identical dilution procedures at controlled stable ambient conditions in the laboratory.

PAAS-4$\lambda$ measurements using monodisperse Nigrosin particles are compared in Fig. 4 to verify the $NO_2$ calibration. For this, Nigrosin dye (Thermo Fisher Scientific Inc., CAS 8005-03-6) is dissolved in water and is dispersed, dried, and size segregated using a differential mobility analyzer (DMA model 3080; TSI Inc., USA). Three sizes were selected and analyzed by PAAS-

4$\lambda$ with mobility equivalent diameters of 150 nm, 200 nm, and 250 nm. It is well known that the residual Nigrosin particles from this aerosolization procedure are spherical (e. g. Lack et al., 2006). With the refractive index of Nigrosin deduced by Bluvshtein et al. (2017) from ellipsometric measurements on thin film samples, the expected spectral absorption cross section of these monodisperse particles can be calculated using Mie theory. A comparison of the measured and calculated spectral absorption cross sections are given in Fig. S7 in the Supplement. A condensation particle counter (CPC, model 3775; TSI Inc.,

USA) was operated parallel to the PAAS-4$\lambda$ behind the DMA to get the particle number concentration of the size-segregated samples. The deduced cell constant $C_{\text{cell}}^{\text{Nigr.}} = 7477$ V·m·W$^{-1}$ is in good agreement with the above $NO_2$ deduced constant, given the elevated uncertainty in the Nigrosin calibration procedure. This uncertainty is due to the combined uncertainties in the particle size selection and the particle number concentration measurement and is expressed by the elevated relative uncertainty of 4% deduced from the regression analysis. An instrument accuracy of about 10% can be estimated from the

two independent calibrations, even though the Nigrosin calibration procedure has not been fully optimized yet. As detailed in Bluvshtein et al. (2017) it is very likely that the remaining discrepancy between the two calibration methods is due the uncertainty in the refractive index of Nigrosin used in the Mie calculations.

### 3.2 Instrument Performance

The detection limit of PAS and filter-based light absorption instruments are usually characterized by performing an Allan

variance or deviation analysis (e. g. Fischer and Smith, 2018; Asmi et al., 2021). Although the Allan deviation analysis is primarily used to define the maximum averaging time that can be deployed without averaging over a significant signal drift, i.e., the characteristic instrument drift stability, the Allan deviation for a specific averaging time also equals the $1\sigma$ detection limit of the instrument in case the signal noise shows white noise characteristics (Asmi et al., 2021). To characterize the PAAS-4$\lambda$ stability and detection limit, particle-filtered laboratory air need to be sampeled over a long period of time (e.g.

40 h in Fig. 5). As can be seen in Fig. 5 the PAAS-4$\lambda$ drift stability is typically between 1000 and 3000 s, when the $1\sigma$ Allan deviation starts to deviate from the $1/\sqrt{\tau}$ white noise characteristic decrease with averaging time $\tau$. This means that after about 30 min measurement time, the instrument background, i.e., particle-filtered ambient air, should be measured to get a drift-free background that corresponds to the aerosol sample period in between. An ultimate $1\sigma$ detection limit around or

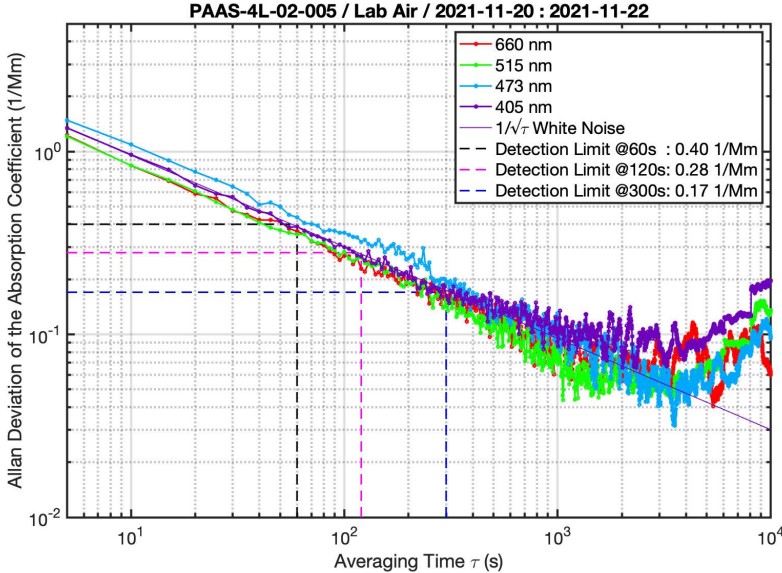

**Figure 5.** Allan deviation analysis of a 40 h background measurement. The instrument sampled particle-filtered laboratory air with a basic averaging time of 5 s per laser wavelength. A white noise characteristic slope is plotted for the 405 nm wavelength (thin purple line). Signal drift starts between 1000 to 3000 s averaging time resulting in an ultimate detection limit of about $0.1 \ \mathrm{Mm}^{-1}$. Detection limits for 60 s, 120 s, and 300 s are indicated by black, magenta, and blue dashed lines, respectively.

below $0.1 \ \mathrm{Mm}^{-1}$ can be deduced from the Allan deviation analysis in Fig. 5. Interestingly, the characteristic PAAS-4λ Allan

deviation plot shown in Fig. 5 is very similar to the corresponding plot shown in Fig. 2 of Fischer and Smith (2018) for a state-of-the-art four-wavelength PAS system with a single PA cell (MultiPAS-IV). This is remarkable given that the MultiPAS-IV uses a multipass mirror set-up around the PA cell to increase the laser power inside the cell by 30 to 56 times, while the PAAS-4λ works with just one laser pass. One can speculate that a PAS optical set-up with just one laser pass through the cell is more stable and less prone to temperature or vibration-dependent alignment drifts compared to a system with an additional

multipass optical cell.

As already mentioned, the Allan deviation analysis is useful to deduce the signal drift stability and the ultimate detection limit, but it can also be used to estimate the detection limits for other, more realistic averaging times. This is especially important for sequentially measuring systems, like the PAAS-4λ, where the actual averaging time is the sample or background period divided by the number of lasers in the system. For long-term field measurements, a typical sample period is 20 min followed

by a particle-filtered background period of 4 min that is sandwiched between two cell flushing periods of 3 min each. Thus, one sample-filter cycle has a total duration of 30 min. This results in effective averaging times per laser of 5 min for the sample period and 1 min for the background period. According to Fig. 5 the $1\sigma$ detection limits for 1 min and 5 min averaging times are $0.4 \ \mathrm{Mm}^{-1}$ and $0.2 \ \mathrm{Mm}^{-1}$, respectively. One can assume that in such a measurement procedure the larger detection limit of the background period will define the overall limit. To verify this assumption the background data of Fig. 5 were reanalyzed





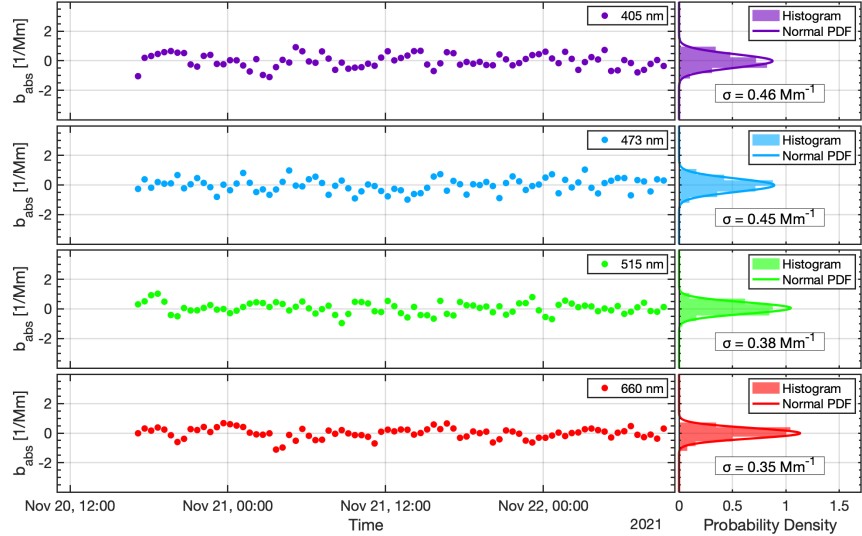

**Figure 6.** Background data of Fig. 5 reanalyzed to simulate a typical measurement sequence where the sample measurements are background corrected every 30 min (left panels). Averaging times are 5 min and 1 min per laser wavelength in the sample and background periods, respectively. Histograms of the background corrected data and corresponding $1\sigma$ deviations are given in the right panels. See text for details.

assuming such a measurement procedure. The 40 h background data are subdivided in alternating sample-filter cycles each consisting of a 20 min "sample" period and a 4 min "background" period that is sandwiched by two 3 min "flushing" periods. The data in each "sample" and "background" period are then averaged resulting in 5 min and 1 min laser averages, respectively. Finally, the averaged data of two "background" periods before and after a "sample" period are averaged and subtracted from the corresponding averaged "sample" data to calculate the background corrected data; the same procedure that is applied to the field data in Sec. 4. The result of this analysis is shown in Figure 6. By comparing Figs. 5 and 6 it is clear that the $1\sigma$ detection limit that is expected for background corrected 20 min sample data in 30 min sample-background cycle is defined by the shorter background periods as the $1\sigma$ values given in Fig. 6 match the 60 s detection limit indicated in Fig. 5.

## 4 Measurement Examples from a Remote Field Site

In the following data from long-term PAAS-4λ measurements at an unattended remote monitoring station is presented. The data is collected at an air quality monitoring station located in the Finnish Artic on top of the Sammaltunturi fell (67°58'N, 24°7'E; 560 m a.s.l.) about 140 km north of the Arctic circle. The station, referred to as "Pallas" from now on, belongs to the Pallas–Sodankylä Atmosphere-Ecosystem Supersite Facility operated by the Finnish Meteorological Institute also as part of the Aerosols, Clouds, and Trace gases Research InfraStructure (ACTRIS). Details of the site and the ongoing measurements can be found in Lohila et al. (2015). The station usually resides in relatively clean Arctic background air with monthly median





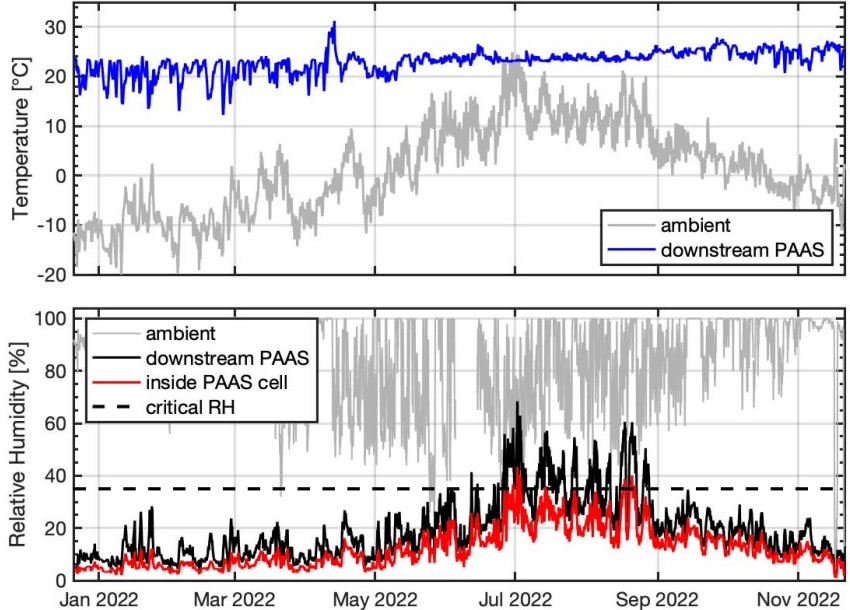

**Figure 7.** Variation of the aerosol temperature and relative humidity (RH) measured downstream the PAAS-4$\lambda$ optics unit inside the flow unit (blue and black lines, respectively). Ambient temperature and RH measured outside the Pallas station are also shown (grey lines). The RH measurement downstream the PA cell is translated to the elevated temperature conditions inside the optics unit by assuming a fixed PA cell temperature of 30 °C (red line). The critical RH of 35 % given by Langridge et al. (2013) for unbiased PAS measurements is indicated by the dashed black line.

BC mass concentrations below $50 \, \mathrm{ng/m^3}$ (Asmi et al., 2021). Only from time to time the station is influenced by long-range transported anthropogenic pollution from central and southern Europe and from biomass burning aerosol emitted by wildfires in northern Eurasia (Hyvärinen et al., 2011). The location of the Pallas station in conjunction with its unattended operation makes it ideal for testing novel monitoring instruments. PAAS-4$\lambda$ measurements at Pallas are ongoing since December 2021.

    Ambient aerosol and gas compositions are continuously monitored at Pallas by a suite of permanently installed instruments,
from which the filter-based absorption photometers Aethalometer® (Magee Scientific Co., USA, model A33) and the MAAP are selected here for the intercomparison of $b_{\mathrm{abs}}(\lambda)$ with PAAS-4$\lambda$ measurements. Further, the PAAS-4$\lambda$ is also compared against a COSMOS instrument that is deployed at Pallas since 2019. COSMOS is a filter-based absorption photometer with a specific inlet that is heated to 300°C to remove volatile light scattering particles and BC coatings from the sampled aerosol. Therefore, the filter-based light attenuation measured by COSMOS has a direct and fixed correlation with the BC mass concen-
tration ($M_{\mathrm{BC}}$) with no cross-sensitivity to light scattering aerosol mass. This has been demonstrated by intercomparison studies with a single-particle soot photometer (SP2, Droplet MeasurementTechnologies, Longmont, CO, USA) in the laboratory as well as at different field sites. Details of the COSMOS instrument can be found in Ohata et al. (2021) and references therein.



The AE33, COSMOS, and PAAS-4$\lambda$ instruments sample at a main inlet which is a total aerosol inlet designed to collect both aerosol particles and the cloud droplets during the station in-cloud periods. Of the total volumetric flow through the total aerosol inlet of $40\,\mathrm{L\,min^{-1}}$ a fraction of $16.5\,\mathrm{L\,min^{-1}}$ is dried with Nafion drier (MD-700, Perma Pure LLC) and divided between different instruments by a sample manifold consisting of $6\,\mathrm{mm}$ outer diameter exit tubes. Conductive silicon tubing with an inner diameter of $4.8\,\mathrm{mm}$ is used to connect the instruments to the manifold. The MAAP instrument is connected to a separate, size-selective interstitial inlet equipped with a $PM_{2.5}$ sampling head and a separate Nafion drier (MD-700, Perma Pure LLC). A difference in aerosol sampling between the total and the interstitial inlet for absorbing aerosol is significant mainly during the in-cloud periods which are therefore separated in data (Hyvärinen et al., 2011). Sampling flows are set to 5.0, 0.7, 10.0, and $1.0\,\mathrm{L\,min^{-1}}$ for the AE33, COSMOS, MAAP, and PAAS-4$\lambda$, respectively. COSMOS uses a cyclone impactor with a cut size of $1\,\mathrm{\mu m}$ to remove supermicron-sized particles from the aerosol. All other instrument do not use an additional impactor.

### 4.1 Instrument Long-Term Performance

**Table 1.** Overview of data gaps and the resulting data coverage during eleven months of unattended operation of PAAS-4$\lambda$ at Pallas, Finland.

| Date | Duration | Reason for data gap | Type |
|---|---|---|---|
| 2022-01-18 | 05:56:38 | Forced operating system update | Unintended |
| 2022-02-11 | 01:39:10 | Forced operating system update | Unintended |
| 2022-07-05 | 40:53:23 | Power outage at the Pallas station | Unintended |
| 2022-08-17 | 01:32:51 | Maintenance and on-site calibration | Intended |
| 2022-09-21 | 19:30:57 | Forced operating system update | Unintended |
| Total gap duration | 68 h | Excluding duration of intended data gaps | |
| Total deployment duration | 8040 h | December 21, 2021 to November 21, 2022 | |
| Data coverage | 99.15% | | |

Eleven months of PAAS-4$\lambda$ data from the Pallas station is used to analyse the instrument long-term performance. This data covers the period from December 21, 2021 to November 21, 2022. During this period the collected data covers 99% of the time (Tab. 1). The remaining unintended data gaps are due to forced operating system updates and a station power outage that all resulted in a forced reboot of the instrument panel computer. None of the data gaps are due to a failure of one of the instrument components like laser, electronics, or flow system components, which clearly demonstrates the technology readiness of PAAS-4$\lambda$ for the unattended long-term operation at remote air quality stations. The relatively long data gaps from the unintended computer reboots in Tab. 1 rather results from the unawareness of the remote user than being evoked by the incident itself. In any of these cases the data acquisition could be restarted via remote access without the attendance of station personnel. An automated data acquisition restart after a computer reboot will be implemented in future long-term deployments.





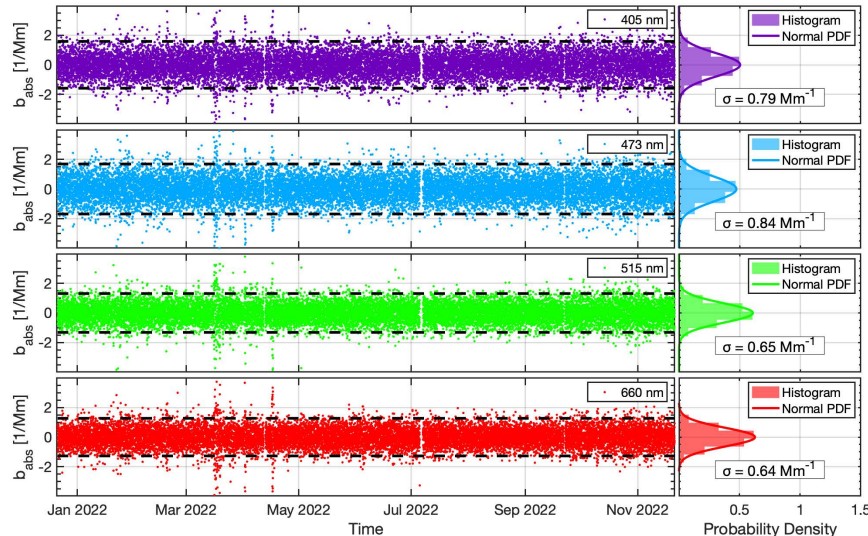

**Figure 8.** Difference in the two filtered air (background) measurements of all filter-sample-filter cycles that were conducted over a period of eleven months during the unattended PAAS-4$\lambda$ deployment at Sammaltunturi. Each data point reflects the difference between two consecutive 1 minute averaged filtered air measurements that are 30 min apart. The dashed horizontal lines represent the $\pm 2\sigma$ detection limit.

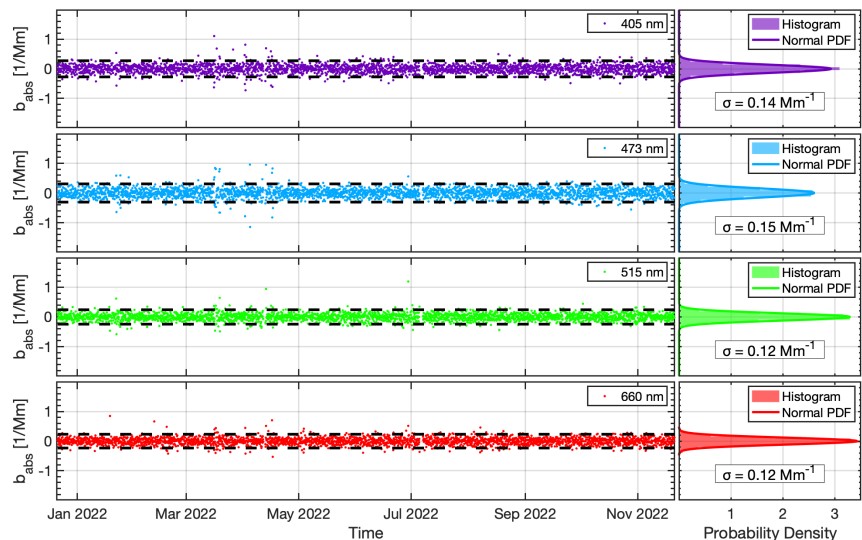

**Figure 9.** Same as Fig. (8) but averaged over three hours.





Figure 7 shows the temperature and relative humidity (RH) variations of the sampled aerosol during this deployment time. The PAAS-4$\lambda$ temperature and RH sensor (E+E Elektronik, Austria; model EE23-T6) is located downstream the PA cell in the flow unit. As mentioned in Sec. 2.1 the dissipated heat from the lasers results in a stabilized temperature inside the optics housing of about 30°C. The PA cell metal body is adapting to this temperature within a warm-up time of about one hour. The RH inside the photoacoustic cell can then be translated from the RH and temperature of the aerosol measured outside the optics housing downstream the PA cell. Figure 7 clearly shows that most of the time the aerosol inside the PA cell has an RH below 30%. Only during the summer months RH was elevated but generally limited to values below 35%, with four days in July where RH reached 40%. This is important as the photoacoustic signal starts to show significant low biases of more than 10% for RH values larger than a critical RH of about 35 % (Langridge et al., 2013).

To verify the detection limit, which has been deduced from the laboratory data shown in Sec. 3.2, for the long-term operation at Pallas, the particle-filtered air (background) data from Pallas is analyzed. The background correction procedure that is applied to the PAAS-4$\lambda$ raw data averages two consecutive one minute averaged background measurements of a filter-sample-filter cycle and subtract this average from the corresponding sample measurement. One can, therefore, assume that the difference between these two background measurements define the uncertainty in the corresponding sample measurement. Figure 8 depicts the difference between two consecutive background measurements that are 30 min apart over the course of the instrument deployment at Pallas; a total of 16,000 data points. The data is normally distributed around the zero line with 1-$\sigma$ distribution widths that are by a factor of roughly two larger than the corresponding one minute averages of the laboratory Allan analysis shown in Fig. 5 and the corresponding background analysis shwon in Fig. 6. In contrast to the laboratory conditions the conditions in- and outside the Pallas station likely cause higher levels of ambient electromagnetic and sound noise as well as larger fluctuations in the temperature, aerosol relative humidity, and trace gas composition of the sampled air, which jointly impacts the PAAS-4$\lambda$ measurements. Especially the impact of light absorbing trace gases like, e. g., $NO_2$ or $O_3$, that are co-transported with combustion aerosol can be clearly seen in Fig. 8 by the higher background fluctuations in the March to May period when the station was frequently influenced by long-range transported pollution episodes (cf. Sec. 4.2).

As the long-term PAAS-4$\lambda$ measurements at Pallas are basically averaged over three hours before using them in any instrument intercomparison or scientific analysis, the background data of Fig. 8 is averaged over three hours in Fig. 9. This averaging pushes the 1-$\sigma$ distribution widths into a range that agrees with the Allan analysis result shown in Fig. 5. A 2-$\sigma$ detection limit of 0.28, 0.3, 0.24, and 0.24 $\mathrm{Mm^{-1}}$ can therefore be deduced for the 3h averaged PAAS-4$\lambda$ measurements from Pallas for the 405, 473, 515, and 660 $\mathrm{nm}$ wavelength, respectively.

## 4.2 Time Series and Instrument Intercomparisons

The time series of background corrected PAAS-4$\lambda$ absorption coefficients $b_{\mathrm{abs}}(\lambda)$ measured at Pallas during the long-term deployment are shown in Fig 10A. Concurrent measurements of the BC mass concentration $M_{\mathrm{BC}}$ by the COSMOS instrument are shown as well. Both data sets are averaged over three hours. The data reflects the in general very clean Arctic background conditions expected at Pallas with annual averages of BC between 60 and 70 $\mathrm{ng\,m^{-3}}$ (Hyvärinen et al., 2011). Actually, the COSMOS data reveals a lower averaged BC mass concentration of only 22 $\mathrm{ng\,m^{-3}}$ for the measurement period discussed




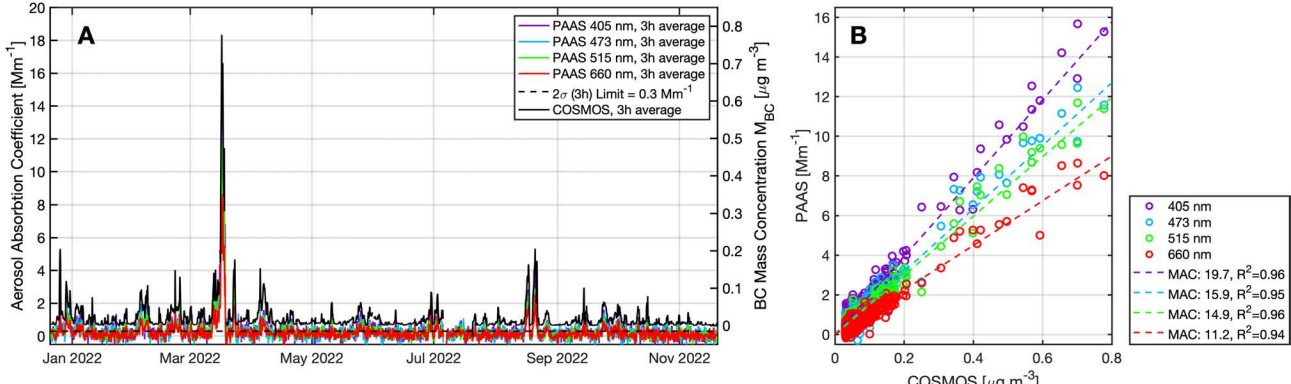

**Figure 10.** Comparison of the PAAS-4$\lambda$ absorption coefficients with concurrent measurements with the COSMOS instrument for a time period of eleven months (A). Correlation analysis of the same data (B), which gives the mass specific absorption cross section (MAC) for each PAAS-4$\lambda$ wavelength given in the legend. Each data set is averaged over three hours.

here. Pollution episodes with higher $b_{abs}(\lambda)$ values of typically 1 to 3 $\mathrm{Mm^{-1}}$, corresponding to $M_{BC}$ values between 50 and 200 $\mathrm{ng\,m^{-3}}$, are clearly distinct from this clean Arctic background. The pollution episodes last for time periods of typically more than 12h and show a higher frequency of occurrence in the winter and spring period compared to summer and autumn. At

Pallas, these episodes can be attributed to polluted air masses that are transported from Central and Eastern Europe to northern Finland (Hyvärinen et al., 2011).

The PAAS-4$\lambda$ absorption coefficients $b_{abs}(\lambda)$ are strongly correlated with the BC mass concentration $M_{BC}$ from COSMOS ($R^2 = 0.94 - 0.96$) resulting in mass specific absorption cross sections (MAC) of 19.7, 15.9, 14.9, and 11.2 $\mathrm{m^2\,g^{-1}}$ for the 405, 473, 515, and 660 $\mathrm{nm}$ wavelengths, respectively (Fig. 10B). Ohata et al. (2021) compared long-term $M_{BC}$ measurements by

COSMOS with concurrent filter-based $b_{abs}$ measurements by Aetholometers, MAAPs, and Particle Soot Absorption Photometers (PSAP) from four Arctic sites including Pallas. They deduced an average MAC of $13 \pm 1.6$ $\mathrm{m^2\,g^{-1}}$ at $\lambda = 550$ nm from all measurements covering a period from 2012 to 2021. Interpolating the MAC(515 nm) = 14.9 $\mathrm{m^2\,g^{-1}}$, deduced from the PAAS-4$\lambda$ versus COSMOS correlation in Fig. 10B, to a wavelength of $\lambda = 550$ nm results in MAC(550 nm) = 13.8 $\mathrm{m^2\,g^{-1}}$, which is in excellent agreement with the Arctic average from Ohata et al. (2021). Here, the Absorption Ångström Exponent

(AAE) of 1.14 deduced from the MAC($\lambda$) of Fig. 10B is used for the interpolation. Further, interpolating the MAC(660 nm) of 11.2 $\mathrm{m^2\,g^{-1}}$ from PAAS-4$\lambda$ to the MAAP wavelength of $\lambda = 637$ nm results in a MAC(637 nm) of 11.7 $\mathrm{m^2\,g^{-1}}$, which is 10% lower than the MAC(637 nm)= 13.0 $\mathrm{m^2\,g^{-1}}$ given by Ohata et al. (2021) for the COSMOS versus MAAP correlation from Pallas for the July 2020 to July 2021 time period, but still within their variability range of 27%.

As mentioned in the Introduction (Sec. 1), a common problem of filter-based measurement methods of $b_{abs}(\lambda)$ is that the

raw filter attenuation data $b_{ATN}$ has to be corrected for multiple light scattering and particle loading effects within the filter matrix, which significantly affects their measurement uncertainty. It has been concluded that concurrent measurements with non-intrusive methods like PAS over longer time periods in the field, would be ideal to further investigate these uncertainties




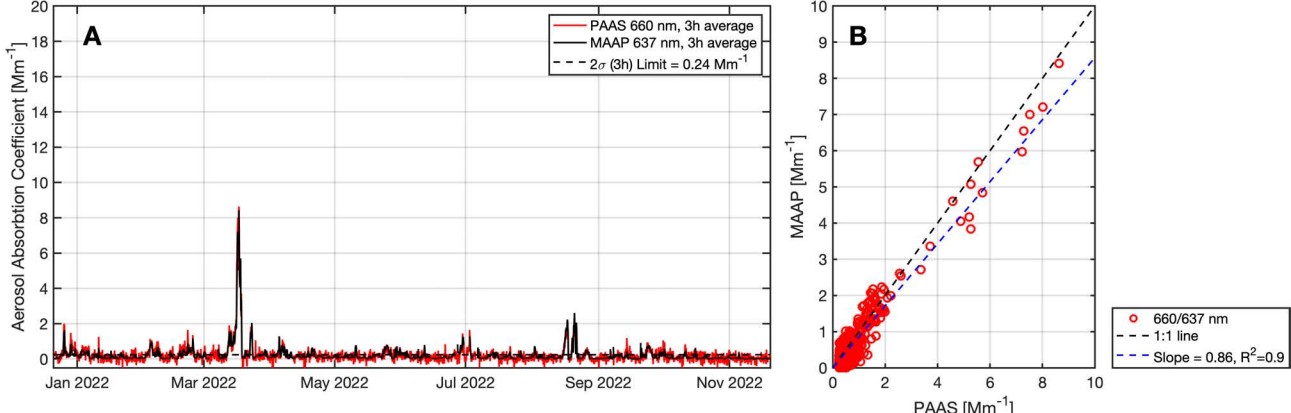

**Figure 11.** Comparison of the PAAS-$4\lambda$ absorption coefficient for 660 nm with concurrent measurements with the MAAP instrument for 637 nm for a time period of eleven months (A). Correlation analysis of the same data (B). Each data set is averaged over three hours.

under target operational conditions. The following paragraphs should therefore briefly emphasize the potential contributions from long-term PAAS-$4\lambda$ deployments in air quality monitoring stations for validating commonly used filter-based instruments
and their correction schemes.

The MAAP instrument is generally considered to be a valid reference for $b_{\mathrm{abs}}$ as it is does not rely on an empiric corrections of the multiple-scattering and loading effects but uses additional measurements of diffusely scattered radiation in the backward hemisphere that allows the determination of $b_{\mathrm{abs}}$ analytically within a radiative transfer model of the particle-loaded filter (Petzold and Schönlinner, 2004). The intercomparison of MAAP with PAAS-$4\lambda$ for the eleven months depolyment period of
PAAS-$4\lambda$ at Pallas verifies this consideration (Fig. 11). The two data sets are well correlated with a correlation coefficient of $R^2 = 0.9$. The regression slope of $0.86$ is within the $\sim 20\%$ combined accuracy of the two instruments, i.e., 10% for PAAS-$4\lambda$ and 12% for MAAP.

Aethalometers® are the most widely used filter-based instruments and are, therefore, most intensively characterized and developed. The latest instrument version AE33 uses a dual spot approach to get an online compensation of the filter loading
effect (Drinovec et al., 2015). Figure 12 shows the intercomparison of AE33 with PAAS-$4\lambda$ over the period of use of PAAS-$4\lambda$ at Pallas so far. Only those wavelengths have been selected from the available seven wavelengths of AE33 that have the closest match with the wavelengths of the PAAS-$4\lambda$ unit deployed at Pallas. The two data sets are well correlated with correlation coefficients $R^2$ between 0.89 and 0.92. The regression slope equals the multiple-scattering correction factor $C$ assuming that $b_{\mathrm{abs}}$ from PAAS-$4\lambda$ represents the unbiased reference. Correction factors of 2.5, 2.4, 2.3, and 2.1 are determined
for the 405 nm/370 nm, 473 nm/470 nm, 515 nm/520 nm, and 660 nm wavelength pairs, respectively. The factor $C = 2.1$ at 660 nm is in a very good agreement with the factors $2.13 \pm 0.57$ for pure Diesel soot and $2.29 \pm 1.36$ for Diesel soot externally mixed with Ammonium Sulphate particles determined by Weingartner et al. (2003) for the same wavelength of the AE30 Aethalometer®. Their data was acquired in aerosol chamber experiments with the difference of spectral extinction and total scattering measurements being the reference for the absorption coefficient (Schnaiter et al., 2003). Further, the $C(660\,\mathrm{nm}) =$



2.1 factor also fits the broad $C(637 \text{ nm})$ factor range of $1.99 - 2.78$ given in Yus-Díez et al. (2021) for AE33 versus MAAP
correlation analyses of data from different urban and regional background as well as mountaintop sites.

The correction factors $C$ of Fig. 12 also indicates a weak negative wavelength dependence with larger values for shorter
wavelengths, which is an interesting observation as it contradicts the current assumption of no or a rather positive dependence
(Yus-Díez et al., 2021). A first statistical analysis of the individual 3h-averaged $C(\lambda)$ factors confirms the wavelength trend
found in the regression analysis with differences in the median $C(\lambda)$ factors that are statistically significant according to the
Wilcoxon rank sum test. However, further detailed analyses of the Pallas data set is necessary to strengthen this observation
also in relation to the actual aerosol composition (i.e., the single scattering albedo). Further, as long-range transported pollution
episodes are nicely contrasting the clean Arctic background with absorption coefficients around the detection limit of PAAS-
$4\lambda$, these episodes can be specifically analysed in terms of instrument intercomparisons. An example of such an analysis is
given in the Supplement (Fig. S9).

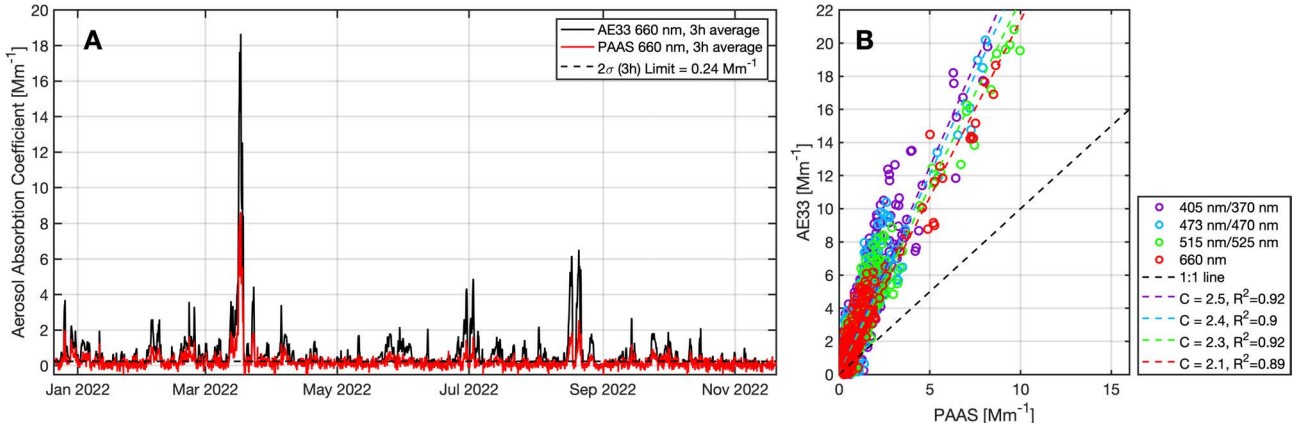

**Figure 12.** Comparison of the PAAS-$4\lambda$ absorption coefficient for 660 nm with concurrent measurements of the same wavelength with the
Aethalometer® (AE33) instrument for a time period of eleven months (A). Correlation analysis of the same data but now for four wavelength
pairs (PAAS/AE33) (B). Each data set is averaged over three hours.

## 5   Conclusions

Photoacoustic aerosol spectroscopy (PAS) is one of the non-intrusive methods to measure the spectral absorption coefficient
$b_{\text{abs}}(\lambda)$ of the atmospheric aerosol that comprises fine mode combustion particles like black carbon (BC) in varying mass
fractions. It represents an alternative method to the commonly used filter-based absorption photometers like the Aethalometer®,
which are prone to measurement errors that are difficult to correct. These errors are caused by multiple light scattering and
total particle light absorption within the filter matrix that is continuously accumulating particles. Even though PAS instruments
are not affected by particle light scattering or total light absorption by the particles, they are commonly regarded to have less
sensitivity and robustness, and need significantly more user interference to keep them operational compared to filter-based





instruments. Therefore, PAS instruments are usually not considered to be suitable for long-term monitoring tasks in (remote)
air quality monitoring stations.

In this work the commercial four-wavelength Photoacoustic Aerosol Absorption Spectrometer PAAS-4$\lambda$ is introduced that has been specifically designed for the unattended operation in remote air quality monitoring stations. The PAAS-4$\lambda$ combines a compact and robust four-laser beam combiner with a single photoacoustic (PA) cell in a simple optical arrangement. This optical set-up is implemented in a laser class 1 enclosure equipped with a thermal concept using the dissipated heat from
the lasers to thermally stabilize PA cell at about 30°C. Dual-phase lock-in technology in combination with an embedded real-time frequency generator is used to sensitively detect the photoacoustic signal even under elevated electronic and sound noise conditions. A touch panel computer is implemented in the electronics unit of PAAS-4$\lambda$ where the instrument application autonomously operate the instrument through predefined measurement sequences that also allow the control of peripheral components, e.g., for implementing inlet switch cycles or calibration procedures in the measurement sequence.

A calibration procedure using certificated $NO_2$/air mixtures in step-wise dilution sequences is applied, which improves the calibration repeatability resulting in a given instrument precision of 3%. When calibrating with $NO_2$, knowledge of the actual laser emission spectra is crucial to achieve a high instrument accuracy. Therefore, the laser emission spectra are routinely measured in the calibration procedures resulting in a given instrument accuracy of 10% that was verified in separate calibration runs using size-segregated Nigrosin particles. The Allan analysis of the PAAS-4$\lambda$ background signal reveals a very good
ultimate $1\sigma$ detection limit of 0.1 $\mathrm{Mm}^{-1}$, which is comparable to other non-commercial state-of-the-art multi-wavelength PAS instruments (e.g. Fischer and Smith, 2018). However, under usual operational settings with particle-free background measurements every 30 min, the shorter background averaging time defines the detection limit. The operational $1\sigma$ detection limit is deduced in the range between 0.35 and 0.46 $\mathrm{Mm}^{-1}$, again in good agreement with the Fischer and Smith (2018) instrument.

To demonstrate the instrument performance under target operational conditions, a PAAS-4$\lambda$ unit has been installed in December 2021 at the remote air quality monitoring station Pallas located in Finland about 140 km north of the Arctic circle. The instrument is continuously acquiring data since then with only very minor attendance from local station personnel. During the deployment so far the instrument worked very reliably with a data coverage of more than 99%. By analysing the background data that was captured every 30 min, $1\sigma$ detection limits between 0.64 and 0.79 $\mathrm{Mm}^{-1}$ are deduced that are higher than those
from the laboratory. Here, the variability of absorbing trace gases in the sampled air might cause at least part of these higher limits. Averaging the data over 3h reduce the detection limits to the range from 0.12 to 0.15 $\mathrm{Mm}^{-1}$.

Instrument intercomparisons between PAAS-4$\lambda$ and the filter-based photometers MAAP, AE33 as well as the COSMOS BC monitor are presented for the eleven-month deployment period at Pallas. In any case there is a strong correlation observed between PAAS-4$\lambda$ and these filter-based instruments. The correlation between PAAS-4$\lambda$ and the BC mass concentration from
COSMOS revealed mass-specific absorption cross sections (MAC) of 11.2, 14.9. 15.9, and 19.7 $\mathrm{m^2\,g^{-1}}$, for the 660, 515, 473, and 405 nm wavelengths, respectively. The single-wavelength MAAP photometer is frequently used as a reference instrument for the aerosol absorption coefficient $b_{abs}$(637 nm), which is confirmed in the correlation analysis with $b_{abs}$(660 nm) from PAAS-4$\lambda$, giving a linear regression slope of 0.86 that is within the combined accuracy of both instruments. Finally, the



intercomparison of PAAS-4$\lambda$ with the Athalometer® AE33 results in multiple scattering correction factors $C(\lambda)$ of 2.1, 2.3, 2.4, and 2.5 for the 660 nm, 515 nm/520 nm, 473 nm/470 nm, and 405 nm/370 nm wavelength pairs, respectively.

The time series and instrument intercomparisons presented in this work are intended to give first insights into the applicability and value of PAAS-4$\lambda$ for long-term aerosol monitoring purposes in (remote) air quality stations. Future analyses of such data will help to better understand the contribution of light absorbing particles to the aerosol composition, sources, atmospheric processing, long term trends, and the role of combustion aerosol for climate forcing. Furthermore, this study has demonstrated
for the first time the utility of the PAS technology for long-term unattended monitoring operations. This makes the technology a noteworthy candidate for the correct quantification of light absorbing particles and their spectral absorption properties in monitoring networks alongside commonly used filter-based methods.

*Data availability.* Upon manuscript acceptance, the PAAS-4$\lambda$, AE33, MAAP, and COSMOS data used for the instrument intercomparisons will be archived in the KITopen repository, the central Open Access publication platform of KIT (contact: KITopen@bibliothek.kit.edu). A
citable persistent identifier (DOI) number will be assigned to the data.

*Author contributions.* MS lead the PAAS-4$\lambda$ development and the installation at Pallas as part of the ATMO-ACCESS project. MS analyzed the PAAS-4$\lambda$, AE33, and MAAP data with the help of AH, EA, HS, and EJ. SO and YK provided the COSMOS data. MS calibrated and characterized the PAAS-4$\lambda$ instrument with help of CL. All were contributing to the interpretation of the results. MS wrote the manuscript with help of all co-authors.

*Competing interests.* Martin Schnaiter and Emma Järvinen are members of schnaiTEC GmbH, the PAAS-4$\lambda$ manufacturer. Martin Schnaiter is part-time employed by schnaiTEC GmbH.

*Acknowledgements.* This work was supported by the Research Program "Changing Earth - Sustaining our Future" of the German Helmholtz Association. EJ is supported by the Helmholtz Association's Initiative and Networking Fund (grant agreement no. VH-NG-1531). EA, HS and AH were supported by the ACCC Flagship of the Academy of Finland under the grant number 337552. The deployment of PAAS-4$\lambda$
at the Pallas–Sodankylä Atmosphere-Ecosystem Supersite Facility is part of a project that is supported by the European Commission under the Horizon 2020 – Research and Innovation Framework Programme, H2020-INFRAIA-2020-1, Grant Agreement number: 101008004. The COSMOS measurements at Pallas are supported by the Japanese Ministry of Education, Culture, Sports, Science and Technology (MEXT); Japan Society for the Promotion of Science KAKENHI Grants (JP20H00638); the Arctic Challenge for Sustainability II (ArCS II) project (JPMXD1420318865); and the Environment Research and Technology Development Funds (JPMEERF20202003) of the Environmental
Restoration and Conservation Agency of Japan.



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
