# Peer review of "The Four-Wavelength Photoacoustic Aerosol Absorption Spectrometer PAAS-4 $\lambda$"

_Atmospheric Measurement Techniques, 2022_

## Referee Comment (RC1)

In this work, the authors describe in detail a photoacoustic aerosol absorption spectrometer PAAS-4λ and analyze its performance, while using it for 11 months of continuous testing in the Arctic. This work is very interesting.

I recommend this manuscript to publish on Atmospheric Measurement Techniques after revision. I have some comments on this manuscript as below:

(1) The authors mention on page 10 that "A correction factor of 1.3 is applied in Fig. 4 for the 405 nm measurements", how did the authors calculate this correction factor?

(2) Why is there no calibration result for the 473 nm band in Figure 4?

(3 Why is the laser emission spectra information in the 473 nm band not shown in Figure S5? The same problem appears in Figure S6, Figure S7, Figure S8.

---

## Author Comment (AC1)

**Answer to Anonymous Referee #2**

**Thank you for your positive evaluation of our preprint and the helpful comments. Below we address your individual comments and describe the corresponding changes to the revised manuscript version. For the sake of clarity our answers are given in bold.**

Main comments:

(1) The authors mention on page 10 that "A correction factor of 1.3 is applied in Fig. 4 for the 405 nm measurements", how did the authors calculate this correction factor?

**As mentioned in the preprint (page 10, lines 243-245) the photolysis of $NO_2$ at wavelengths below about 420 nm affects the photoacoustic signal generation, which must be corrected when using the 405 nm laser in the calibration of the photoacoustic (PA) system. Two processes need to be considered here:**

**1.) Photolysis refers to the process in which high-energy photons cause the dissociation of $NO_2$ molecules into NO and O. In the wavelength region between the dissociation limit of $NO_2$ ($\lambda$=398 nm) and ~420 nm the quantum yield for the $NO_2$ photolysis rapidly decreases from unity to near zero. The wavelength dependence of the quantum yield has been measured by Roehl et al., 1994, resulting in a value of 0.41 for $\lambda$=405 nm. The actual emission peak was found around $\lambda$=404.8 nm for the PAAS unit PAAS-3L-01-003 used in this calibration (Fig. S5), so we estimated the yield from the adjacent yields for 404 nm and 405 nm given in Roehl et al. (1994). This means that 41% of the absorbed photons are used in the dissociation process and, therefore, are lost for the PA signal. Hence, the PA signal is reduced by a factor of 0.59, which can be accounted for in the calibration procedure by multiplying the 405 nm PA signal with a correction factor of 1/0.59=1.69.**

**2.) One key reaction that can occur after the dissociation of $NO_2$ is the formation of ozone ($O_3$), where the oxygen atom that is released by the $NO_2$ dissociation reacts with an oxygen molecule to form an ozone molecule ($O+O_2->O_3$). This reaction is exothermic, meaning that it releases 143 kJ/mol (=1.48 eV) of heat to the surrounding gas, which in turn contributes to the generation of the PA signal.**

**Consequently, about 20% of the initial 405 nm (3.06 eV) photon energy is gained back by the ozone formation:**

**1.48 eV * 0.41 = 0.61 eV -> 0.61 eV/3.06 eV = 0.20.**

**This reduces the loss of absorbed photon energy that is not available for generating the PA signal due to NO2 photolysis to a net loss of 21% (41% - 20%). As a result, a correction factor of approximately 1.3 must be applied to the 405 nm PA signal.**

**It is important to note that the contribution of the ozone formation to the PA signal has not been sufficiently investigated. Correction factors between 1.29 (Tien, Moosmüller, Arnott, 2013) and 1.56 (Nakayama et al., 2015) were reported for the 405 nm wavelength using the $b_{abs}$ vs. $b_{ext}$ method where the PA cell is filled with a very high concentration of $NO_2$ (typically several 100 ppm). These authors attribute the correction factors to the photodissociation yield only. However, a detailed analysis of the $NO_2$ photolysis requires the knowledge of the actual laser emission spectrum, which – to our knowledge – has not been measured in those studies.**

**After correcting the 405 nm calibration, the cell constants derived separately for the 405, 515, and 660 nm wavelengths agreed with the total constant by 0.2%, 6%, and 0.04%, respectively.**

**We added the following paragraph to the revised manuscript to explain our calculations more detailed:**

**"According to the data presented in Roehl et al. (1994), the $NO_2$ photolysis quantum yield is 0.41 for the peak emission wavelength of 404.8 nm, as deduced for the PAAS-4λ unit used in this calibration (PAAS-3L-01-003, Fig. S5). This means that only 59% of the absorbed photon energy is transferred into the PA signal. However, after the dissociation of $NO_2$, the released oxygen atoms can react with oxygen molecules to form ozone. This reaction is exothermic and therefore has the potential to contribute to the generation of the PA signal. The released heat of 143 kJ/mol accounts for about 20% of the initially absorbed photon energy, increasing the energy fraction available for the generation of the PA signal to 79%. Based on this calculation a correction factor of 1.3 is applied in Fig. 4 for the 405 nm measurements, which is within the range of correction factors between 1.29 to 1.56 reported elsewhere for the same laser wavelength (Tian et al., 2013; Nakayama et al., 2015)."**

(2) Why is there no calibration result for the 473 nm band in Figure 4?

**As detailed in Section 2.1.1 on the "Laser System," the PAAS-4λ instrument family utilize a laser beam combiner capable of accommodating up to four lasers with user-defined wavelengths. For the very detailed calibration study presented in the manuscript, a unit (S/N PAAS-3L-01-003) was used that contained only three lasers with 405, 515, and 660 nm nominal emission wavelengths. That is why there is no 473 nm calibration shown in Fig. 4. We included the S/N of this unit into the caption of Fig. 4 to make this clearer.**

(3) Why is the laser emission spectra information in the 473 nm band not shown in Figure S5? The same problem appears in Figure S6, Figure S7, Figure S8.

This has the same reason as detailed in the previous answer. The unit with S/N PAAS-3L-01-003 that was used for the laboratory calibration study shown in the manuscript hosts only three lasers with 405, 515, and 660 nm nominal emission wavelengths. That is why there is no 473 nm data shown in Fig. S5 (bottom), S6 and S7. The unit PAAS-4L-02-005 shown in Fig. S5 (top) hosts a 473 nm laser, which has been characterized in the same way as the other lasers (see figure below). We have excluded this graph from Fig. S5 since its primary objective is to illustrate how lasers emitting at the same nominal wavelength may have spectral shifts, resulting in a noteworthy modification of the laser emission's specific $NO_2$ absorption cross section that directly translates into the calibration constant.

[Figure]

The unit with S/N PAAS-3L-01-002 that was used in the intercomparison in Fig. S8 hosts four lasers emitting at 405, 515, 660, and 785 nm. However, the prototype instrument KITPAAS, which was used in the intercomparison shown in Fig. S8, hosts only three lasers with emission wavelengths of 445, 520, and 658 nm. We changed the captions of Fig. S6, S7, and S8 accordingly.

---

## Author Comment (AC2)

**Answer to Anonymous Referee #1**

**Thank you for your positive evaluation of our preprint and the helpful comments. Below we address your individual comments and describe the corresponding changes to the revised manuscript version. For the sake of clarity our answers are given in bold.**

GENERAL REMARKS

The manuscript presents an exhaustive description of a novel four-wavelength photoacoustic aerosol absorption spectrometer, together with the results of carefully conducted assessment studies of instrument performance. The instrument evaluation is completed by an instrument intercomparison study with filter-based light absorption measurement instruments at a rural background station in Finland. The assessment reports method-characteristic parameters like limit of detection, precision, and accuracy. The study is carefully designed and performed. The presentation of the results is well structured and clear. The manuscript fits well into the scope of the journal and can be accepted for publication, after few technical corrections have been implemented.

SPECIFIC COMMENTS

The only point which may call for clarification is the use of the term "ultimate detection limit". In the manuscript this term is used together with "detection limit". The authors may want to clarify whether the two terms are used synonymously or have a different meaning.

**The terms "ultimate detection limit" and "detection limit" have different meanings here. We use the former in context of the Allan deviation analysis shown in Fig. 5, where the "ultimate detection limit" is defined by the minimum 1-σ deviation of < 0.1 1/Mm that is observed before the drift stability of the instrument increases the deviation for averaging times longer than 1000 to 3000 seconds. Although the "ultimate detection limit" is a characteristic of the instrument, it is of little practical use, as also mentioned by Fischer and Smith (2018).**

**A more accurate indicator is the "(practical) detection limit", which is defined by the actual time sequence of particle-filtered background measurements taken under real operational conditions. This approach provides a more realistic understanding of how the PAAS-4λ measurements are affected in long-term monitoring applications. For the long-term field measurements presented in the manuscript, background measurements are taken every 30 minutes with an averaging time of 1 minute per laser. Analyzing the particle-filtered laboratory data for such a measurement sequence in Fig. 6 gives a 1-σ**

**detection limit of 0.4 1/Mm in a good accordance with the 60s averaging result from the Allan analysis (Fig. 5).**

**We changed Fig. 5 and its caption to clarify the terms "Ultimate Detection Limit" and "Practical Detection Limit". In the manuscript, the latter term is used synonymously with "Detection Limit".**

[Figure]

**Figure 5. Allan deviation analysis of a 40h background measurement. The instrument sampled particle-filtered laboratory air with a basic averaging time of 5 s per laser wavelength. A white noise characteristic slope is plotted for the 405 nm wavelength (thin purple line). Signal drift starts between 1000 to 3000 s averaging time resulting in an ultimate detection limit of less than 0.1 Mm$^{-1}$ for these averaging times. More practical detection limits for averaging times of 60 s, 120 s, and 300 s are indicated by black, magenta, and blue dashed lines, respectively.**

The following minor issues are mostly suggestion for rephrasing for the sake of more clarity or readability.

MINOR ISSUES

Line 15: The expression "measured across the filter thickness" may be replaced by "measured by light transmission through the filter.

**Changed as suggested.**

Line 29: Replace "for a several years data set" by "based on a multi-year data set".

**Changed as suggested.**

Line 43: References might be added for the use of the AAE to separate biomass burning aerosol (Sandradewi et al., 2008; Kirchstetter et al., 2004) and dust (Petzold et al., 2009) from fossil fuel combustion aerosol.

**Added as suggested.**

Line 75: The description of the photoacoustic process reads like it works only for externally mixed particles. Suggestion for rephrasing: "in an aerosol containing light-absorbing particulate matter compounds".

**Changed as suggested.**

Line 93: Suggestion "are presented in more detail".

**Changed as suggested.**

Line 110: Beam ellipticity might be reported as digits with similar decimal places, i.e., 1.1:1.0.

**Changed as suggested.**

Line 220: Rephrase: "measured prior to …"

**Rephrased to: "Therefore, each laser unit's emission spectrum is evaluated using a compact Czerny-Turner CCD spectrometer (CCS100/M, Thorlabs Inc., USA) with a spectral accuracy better than 0.5 nm within the 350 nm to 700 nm spectral range, before being installed into PAAS-4λ."**

Line 299: Shouldn't it read. "… the same procedure is applied to …"?

**Changed as suggested.**

Line 323: Suggestion: "… sample from a main inlet …"

**Changed as suggested.**

Line 340: Please correct: "…long data gaps … rather result from …".

**Corrected.**

Figure 10: Plotting the y-axis with a log scale might better show the good data agreement at the low b_abs values.

**We investigated this suggestion and concluded that plotting graph A of Fig. 10 with logarithmic y-axis does not improve the visibility of the data agreement. However, we found that plotting graph B on double logarithmic scale does improve the visibility of the data agreement. Fig. 10 will be changed accordingly in the revised manuscript.**

[Figure]

Section 5: This section presents more a summary than conclusions. It should be modified or the section header should be adjusted.

**Changed to "Summary"**

Reference list: Most of the journals are given with their full names, but some are referred to with abbreviations. Then, some of the journal names start with capital letters, others don't. This should be made consistent.

**The reference list will be made consistent in the revised manuscript.**

References

Kirchstetter, T. W., Novakov, T., and Hobbs, P. V.: Evidence that the spectral dependence of light absorption by aerosols is affected by organic carbon, Journal of Geophysical Research, 109, D21208, 10.1029/2004JD004999, 2004.

Petzold, A., Rasp, K., Weinzierl, B., Esselborn, M., Hamburger, T., Dörnbrack, A., Kandler, K., Schütz, L., Knippertz, P., Fiebig, M., and Virkkula, A.: Saharan dust refractive index and optical properties from aircraft-based observations during SAMUM 2006, Tellus Series B-Chemical and Physical Meteorology, 61, 118–130, 10.1111/j.1600-0889.2008.00383.x, 2009.

Sandradewi, J., Prevot, A. S. H., Weingartner, E., Schmidhauser, R., Gysel, M., and Baltensperger, U.: A study of wood burning and traffic aerosols in an Alpine valley using a multi-wavelength Aethalometer, Atmospheric Environment, 42, 101-112, 10.1016/j.atmosenv.2007.09.034, 2008.